# Tributyrin alleviates gut microbiota dysbiosis to repair intestinal damage in antibiotic-treated mice

Ning Yang[1☯], Tongtong Lan[1☯], Yisa Han[1], Haifeng Zhao[2], Chuhui Wang[1], Zhen Xu[1], Zhao Chen[1], Meng Tao[1], Hui Li[1], Yang Song[1]*, Xuezhen Ma[3]*

1 Department of Nutrition and Food Hygiene, School of Public Health, College of Medicine, Qingdao University, Qingdao, China, 2 Qingdao Institute of Food and Drug Control, Key Laboratory of Quality Research and Evaluation of Marine Traditional Chinese Medicine, State Medical Products Administration, Qingdao, China, 3 The Affiliated Qingdao Central Hospital of Qingdao University, The Second Affiliated Hospital of Medical College of Qingdao University, Qingdao, China

☯ These authors contributed equally to this work.
* qdsongyang@126.com (YS); 18660229289@126.com (XM)

**Data Availability Statement:** The raw data of 16S rRNA gene libraries generated during this study is publicly available at the Sequence Read Archive (SRA) portal of NCBI under accession number

## Abstract

Tributyrin (TB) is a butyric acid precursor and has a key role in anti-inflammatory and intestinal barrier repair effects by slowly releasing butyric acid. However, its roles in gut microbiota disorder caused by antibiotics remain unclear. Herein, we established an intestinal microbiota disorder model using ceftriaxone sodium via gavage to investigate the effects of different TB doses for restoring gut microbiota and intestinal injury. First, we divided C57BL/6 male mice into two groups: control (NC, n = 8) and experimental (ABx, n = 24) groups, receiving gavage with 0.2 mL normal saline and 400 mg/mL ceftriaxone sodium solution for 7 d (twice a day and the intermediate interval was 6 h), respectively. Then, mice in the ABx group were randomly split into three groups: model (M, 0.2 mL normal saline), low TB group (TL, 0.3 g/kg BW), and high TB group (TH, 3 g/kg BW) for 11 d. We found that TB supplementation alleviated antibiotics-induced weight loss, diarrhea, and intestinal tissue damage. The 16S rRNA sequence analysis showed that TB intervention increased the α diversity of intestinal flora, increased potential short-chain fatty acids (SCFAs)-producing bacteria (such as *Muribaculaceae* and *Bifidobacterium*), and inhibited the relative abundance of potentially pathogenic bacteria (such as *Bacteroidetes* and *Enterococcus*) compared to the M group. TB supplementation reversed the reduction in SCFAs production in antibiotic-treated mice. Additionally, TB downregulated the levels of serum LPS and zonulin, *TNF-α*, *IL-6*, *IL-1β* and *NLRP3* inflammasome-related factors in intestinal tissue and upregulated tight junction proteins (such as *ZO-1* and *Occludin*) and *MUC2*. Overall, the adjustment ability of low-dose TB to the above indexes was stronger than high-dose TB. In conclusion, TB can restore the dysbiosis of gut microbiota, increase SCFAs, suppress inflammation, and ameliorate antibiotic-induced intestinal damage, indicating that TB might be a potential gut microbiota modulator.

PRJNA915186. The rest of the data is in the Supporting information files.

**Funding:** This study was supported by the National Natural Science Foundation of China (81973033). The funders had no role in study design, data collection and analysis, decision to publish, or preparation of the manuscript.

## Introduction

The misuse and overuse of antibiotics are increasing worldwide, especially in China [1]. The abuse of antibiotics is associated with an increase in gastrointestinal tract chronic inflammatory diseases, including chronic diarrhea, diabetes, and inflammatory bowel disease [2]. Increasing evidence has indicated that gut microbiota disorders are associated with pathogenic mechanisms for disease onset [3, 4]. Therefore, potential therapies to remediate gut microbiome disorders need to be studied.

Short-chain fatty acids (SCFAs) are small molecules produced by intestinal flora fermentation [5]. Numerous studies have shown that SCFAs play an anti-inflammatory and intestinal barrier repair role in the gut [6, 7]. Also, the benefits of most intestinal beneficial bacteria are mediated by SCFAs. When the gut microbiota disorders, SCFAs, particularly butyric acid, decrease [8]. Butyric acid is one of the principal sources of energy for gut microbiota. Many *in vivo* and *in vitro* experiments have shown that butyrate plays a significant role in the gut, such as regulating immune and inflammatory responses and repairing intestinal barriers [9–14]. A previous study has shown that sodium butyrate could reduce the development of autoimmune hepatitis by modulating immunomodulatory cell and intestinal barrier function [15]. A recent study has also shown that sodium butyrate supplements could alleviate steatohepatitis caused by a high-fat diet [16]. These diseases have been associated with gut microbiota. However, the roles of butyric acid in gut microbiota disorder caused by antibiotics remain unclear. Therefore, the restorative effect of butyric acid on intestinal flora disorders is worthy of further study. However, butyric acid is a small molecule fatty acid that easily decomposes. On the other hand, tributyrin (TB) is a pre-butyrate drug without an unpleasant odor compared to direct butyrate intervention. Additionally, TB is not broken down by gastric juices and is slowly converted into butyric acid and glycerol in the gut by pancreatic lipases [17]. Hence, TB might work better than direct butyric acid interventions. Different doses of TB might produce different or opposite effects [18]. For example, butyrate can inhibit inflammation and promote intestinal barrier function at low concentrations [19, 20], while high concentrations can promote inflammation and damage intestinal barrier function by inducing apoptosis [19, 21]. Besides, our previous study showed that an appropriate TB dose could inhibit the occurrence and metastasis of colorectal cancer, whereas a high dose did not play a better role. Thus, whether different doses of butyric acid have different effects on gut microbiota needs further investigation. The above evidence suggested that TB, as a butyric acid precursor, might have a key role in restoring intestinal flora disorder by slowly releasing butyric acid.

In the present study, we explored whether TB intervention can repair intestinal damage and the mechanisms caused by antibiotics by improving gut inflammation and gut microbiota imbalance. We also investigated whether different doses affect gut microbiota and intestinal inflammation.

## Materials and methods

### Animals and experimental design

Thirty-two male C57BL/6 mice (18–21 g, 6 weeks old) were provided by SPF BIOTECHNOLOGY Co., Ltd (Beijing, China). The Affiliated Hospital of Qingdao University's Animal Ethics Committee examined and approved the animal study (Approval No. 20211209C57BL/64820220104084). After adaptive feeding for seven days, mice were randomly divided into control (NC, n = 8) and experimental (ABx, n = 24) groups. The NC group received intragastric perfusion with 0.2 mL of normal saline. In contrast, the ABx group received intragastric perfusion with 0.2 mL of 400 mg/mL ceftriaxone sodium (Dalian Meilun Biotech Co., Ltd.

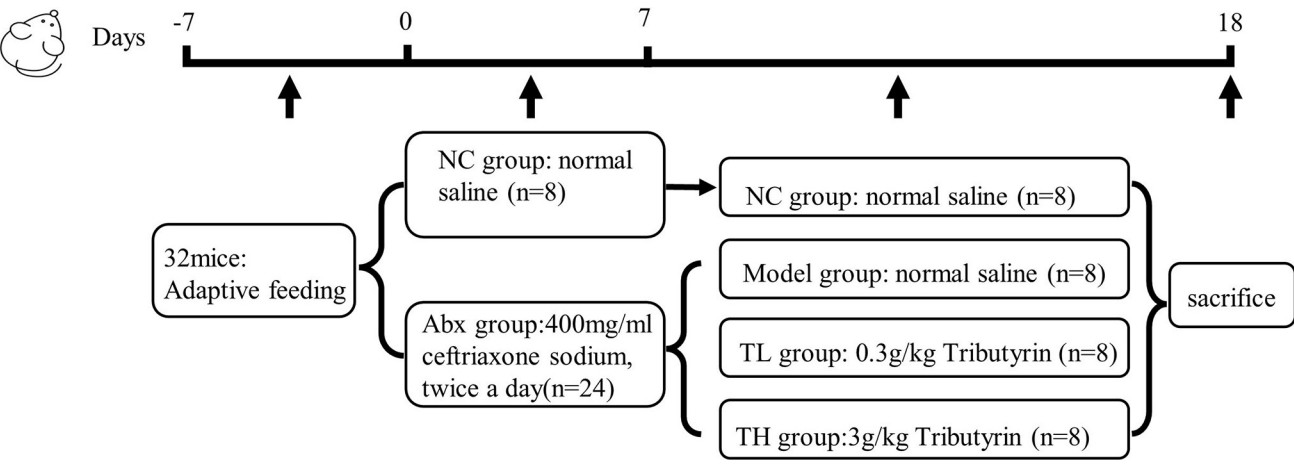

**Fig 1. Study design showing model construction, grouping, and intervention.**

China.) solution for seven consecutive days, twice a day, with intermediate intervals of six hours, resulting in intestinal flora dysregulation [22]. Then, ABx mice were randomly divided into model (M, n = 8), low TB (TL, n = 8), and high TB (TH, n = 8) groups. TL and TH mice were treated with TB solution (98% purity, Macklin Biochemical Co., Ltd. China) at a daily dose of 0.3 and 3g/kg, respectively, for 11 d (The duration of the intervention was based on previous studies [23–26]) (Fig 1). We recorded the body weight, water consumption, and fecal characteristics of mice on alternate days. Fecal traits were divided into five grades [27]: 0, normal; 1, slightly wet; 2, moderate moisture; 3, loose; 4, water sample stool. We collected mice feces on days 7 and 18 and stored them at -80°C. All mice were sacrificed on day 18.

We also measured the cecum weight and the colon length. We calculated the cecal index using the formula: cecum index = [cecum weight (mg)/ mouse weight (g)] [28]. Finally, we collected tissue and blood for subsequent analysis.

## Histological analysis of colon

Colonic tissues from each group were cleaned of feces with PBS before being preserved in 4% paraformaldehyde for histological examination. The paraffin-embedded fixed tissue was divided into 6 μm slices. We produced tissue slices for hematoxylin and eosin (HE) and alcian blue stainings and used a light microscope to examine the colon's histomorphology.

## 16S rRNA gene sequencing analysis

Total genomic DNA from mouse feces was recovered using the OMEGA DNA Kit (Omega Bio-Tek, Norcross, GA, USA). A 0.8% agarose gel electrophoresis was used to assess DNA quality, and Nanodrop was used to measure the DNA. Polymerase chain reaction (PCR) with 338F (5 '- ACTCCTACGGGAGGCAGCA-3') and 806R (5 '- GGACTACHVGGGTWTCT AAT-3') primers was used to amplify the bacterial 16S rRNA, especially factor V3 and V4 DNA regions. Thermal cycling comprised 98°C for 5 min, 25 cycles of 30 s at 98, 53, and 72°C, and a final extension of 5 min at 72°C. PCR products were measured using the Quant-iT Pico-Green dsDNA Assay Kit on a Microplate Reader (BioTek, FLx800), then combined following the amount of information needed for each sample. Libraries were created using the Illumina TruSeq Nano DNA LT Library Prep Kit and examined and sequenced Using the NovaSeq 6000 SP Reagent Kit and Illumina NovaSeq (500 cycles).

## Analysis of SCFAs

Mice feces were weighed (20–40 mg), supplemented with first-level water (Millipore water preparation; note: the whole process was operated on a ten-thousandth scale) to 300 mg, then homogenized with glass beads (60 Hz, 30 + 15 + 30 s). Next, acetone 600 μL was added and swirly mixed, followed by 50% sulfuric acid 30 μL, ether 600 μL, sodium chloride 90 mg, activated carbon 90 mg, swirl mix 2 min, and 0°C frozen centrifugation for 3 min (8000–10000 rpm). The upper organic clear solution was recovered. After 0.22 μm organic needle filtration, the first two drops of the filtrate were discarded, and the filtrate was collected for GC/MSD analysis.

The chromatographic conditions were as follows: the inlet temperature was 230°C, the carrier gas was high purity helium, and the column flow rate was 1.0 mL/min (constant current mode). The initial temperature was 70°C for 1 minute, and the temperature was heated to 110°C at 20°C /min, and then heated to 230°C at 10° C /min at 180°C at 5°C /min for 6 minutes. The mass spectrum conditions are as follows: the ionization mode is the electron bombardment ion source (EI), the electron bombardment energy is 70 eV, the ion source temperature is 230°C, and the electron multiplier voltage is the automatic tuning value. The scanning mode of mass spectrum data is selective ion scanning (SIM). Selected ions: 41, 42, 43, 45, 55, 57, 60, 73, 74, 87, 88, solvent delay time was 4.2 minutes.

## Quantification of blood samples LPS and Zonulin by ELISA

After animal experiments and fasting for 12 h, mice were anesthetized and sacrificed for cervical dislocation after blood series with the eyeball approach. Serum was obtained by centrifugation (Sigma3K15, Germany) at 1500 $g$ and 4°C for 10 min. LPS and zonulin serum concentrations were quantified using ELISA kits (Wuhan Boster organic generation., Ltd, China) following the manufacturer's instructions.

## Real-time quantitative PCR

Total RNA was extracted from colon tissues with TRIzol, then reversely transcribed into cDNAs. Real-time PCR was used to assess mRNA expression using *GAPDH* as an internal reference. The primers (20 μL) used were: *Il6* (for *IL-6*), *tnf-alpha* (for *TNF-α*), *MUC2* (for mucin 2), *Ocln* (for *Occludin*), *Tjp1* (for *ZO-1*), *Nlrp3* (for *NLRP3*), *Casp1* (for *caspase-1*), *Pycard* (for *ASC*), and *Il1b* (for *IL-1β*). Amplifications were conducted at 95°C for 60 s, 1 cycle; 95°C for 10 s, 58°C for 10 s, 40 cycles; 72°C for 15 s, 95°C for 15 s, 60°C for 60 s, 1 cycle. Gene levels were calculated using the $2^{-\Delta\Delta CT}$ approach.

## Western blot

First, colon tissues were homogenized with glass beads (60 Hz, 30 + 30 s), then lysed using the RIPA&PMSF lysis buffer (Beijing solarbio technology & generation Co., Ltd, China). Lysates were centrifuged at 12,000 rpm for 5 min. The supernatant was collected and mixed with a five-fold sodium dodecyl sulfate (SDS) buffer. Samples were separated by 6–12% acrylamide gel electrophoresis and transferred to the polyvinylidene fluoride film. After incubation with primary and secondary antibodies, protein bands were detected by VILBER Fusion FX7 and analyzed by ImageJ.

The following antibodies were used: *ZO-1* Rabbit polyclonal antibody (Cat.No. YN1410, Immunoway), *Occludin* Rabbit polyclonal antibody (Cat.No. 381549, ZENBIO), *NLRP3* Rabbit polyclonal antibody (Cat.No. 381207, ZENBIO), *caspase-1* Rabbit polyclonal antibody (Cat. No. 342947, ZENBIO), *ASC* Rabbit polyclonal antibody (Cat.No. 340097, ZENBIO), *IL-1β* Rabbit polyclonal antibody (Cat.No. TA5103, Abmart), *β-actin* Rabbit polyclonal antibody (Cat.No. AC006, Abclonal).

## Statistical analysis

Data are expressed as means ± standard errors (SE). Statistical analyses were conducted using Graph pad prism 8. Multiple groups were compared by analysis of One-Way Analysis of Variance (One-Way ANOVA) followed by Tukey's multiple comparison test. Statistical significance was set at $P < 0.05$.

## Results

### Effects of TB on body weight, food consumption, water intake, and fecal characteristics of mice

The changes in body weight, food consumption, and water drink in mice of the NC group were kept at a relatively regular level during experiments. After antibiotic treatment, M, TL, and TH mice developed slight to moderate diarrhea signs, lower food consumption, slow frame weight gain, and accelerated water consumption compared to the NC group. Low- and high-dose TB treatment quickly decreased water intake and fecal score, increased food consumption, and recovered body weight gain. By the end of the study, the body weight and diarrhea rating of M mice were not completely recovered ($P < 0.05$, Fig 2A–2D).

### TB alleviated antibiotics-induced colitis in mice

The cecum weight and colon length of M mice increased compared to the NC group. After TB intervention, the cecum of the TL and TH groups was significantly reduced, and the cecal index and colon length were significantly smaller than those of the M group ($P < 0.05$), the

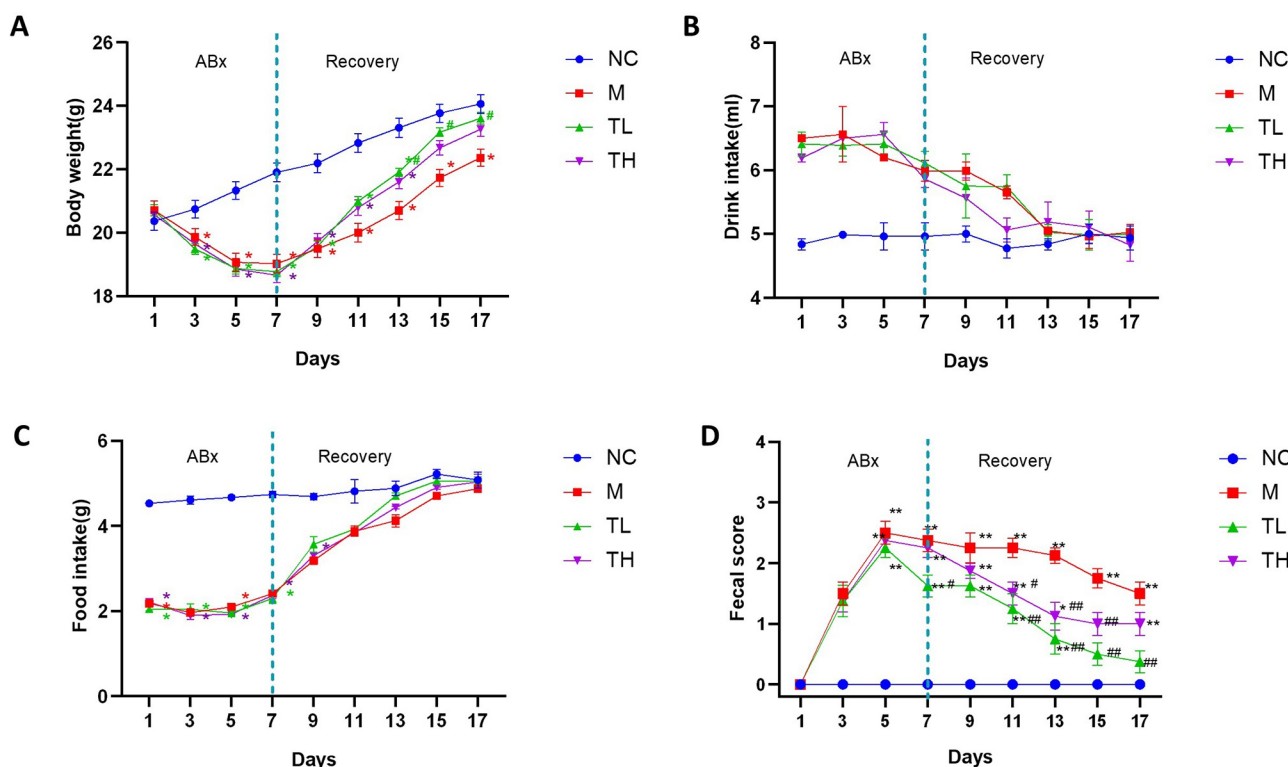

**Fig 2.** Effects of TB treatment on (A) body weight, (B) food intake, (C) water intake, (D) fecal score. NC: control group; M: model group; TL: TB low dose group; TH: TB high dose group. *P<0.05; **P<0.01 as compared with the NC group; #P<0.05, ##P<0.01 as compared with the M group.

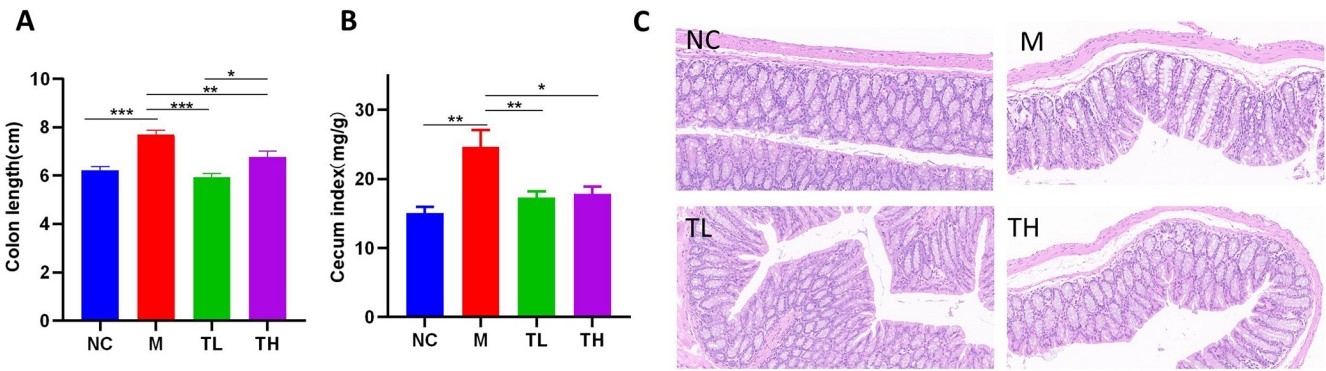

**Fig 3.** TB alleviated antibiotic-induced colitis in mice (A) Cecum index, (B): colon length, (C): HE staining, 400×. NC: control group; M: model group; TL: TB low dose group; TH: TH high dose group. *P<0.05; ** P<0.01; ***P<0.001.

colons of mice in the TH group were longer than those in the TL group (P<0.05, Fig 3A and 3B). Furthermore, the colonic tissues of NC mice were intact, the colonic epithelial cells were closely arranged, and there was no inflammatory cell infiltration. Compared to the NC group, colonic tissue edema occurred in the M group, intestinal mucosa continuity was damaged, epithelial cells were shed, and goblet cells were depleted. Compared to the M group, the colonic tissue edema was significantly improved, and intestinal mucosal continuity was restored in the TL group. The continuity of intestinal mucosa in the colon tissue of the TH group was restored, but there was still edema (Fig 3C).

## TB adjusted antibiotic-induced gut microbiota disorder in mice

### TB improved the diversity and altered the structure of gut microbiota in antibiotic-induced mice.
Among the α diversity indexes, Chao1 and Shannon's indices mainly reflect

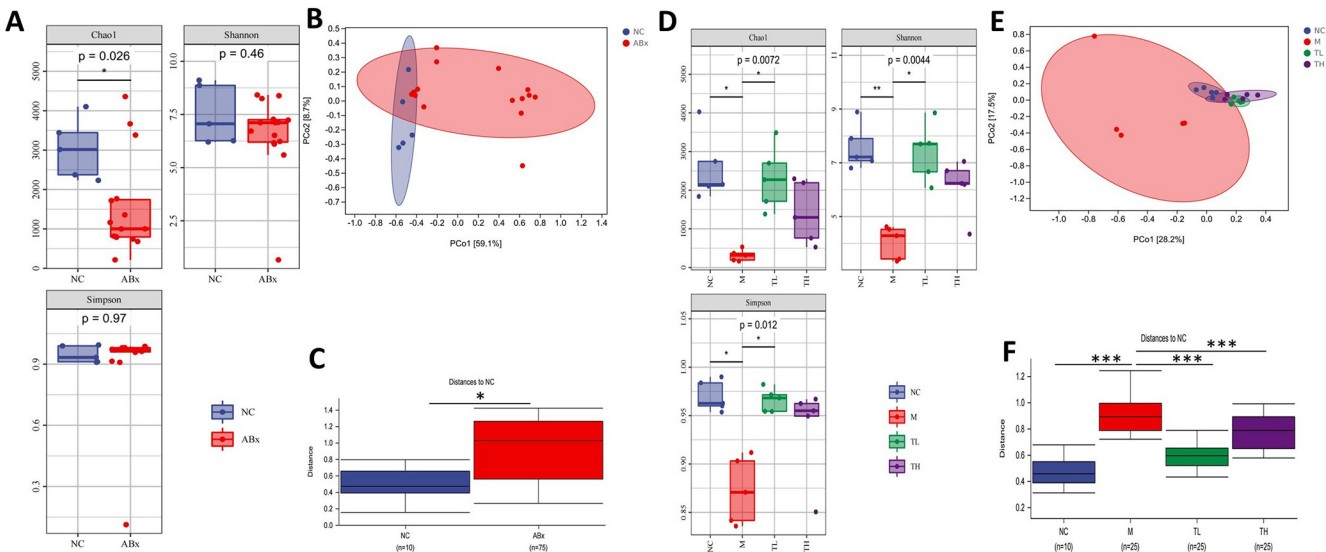

**Fig 4. TB improved the diversity and altered the structure of gut microbiota in antibiotic-induced mice. (A)** On the 7-day, Chao1, Shannon and Simpson indices, **(B)** On the 7-day, Principal coordinate analysis (PCoA) based on weighted unifrac distance, **(C)** On the 7-day, Weighted unifrac distance, **(D)** On the 18-day, Chao1, Shannon and Simpson indices, **(E)** On the 18-day, Principal coordinate analysis (PCoA) based on weighted unifrac distance, **(F)** On the 18-day, Weighted unifrac distance, NC: control group; ABx: ABx group; M: model group; TL: TB low dose group; TH: TB high dose group. *P<0.05; ** P<0.01; ***P<0.001.

the sample community richness and variety, respectively. Compared with the NC group, the Chao1 index of gut microbiota in the ABx group decreased significantly (P<0.05) (Fig 4A). After 11 days of TB intervention, the α diversity index of gut microbiota in the M group was still lower than that in the NC group (P<0.05). Gut microbiota Chao1, Shannon and Simpson indices in the TL group were significantly higher than those in the M group (P<0.05), TH group had the same trend, but the difference was not statistically significant (P>0.05) (Fig 4D).

Then, we used PCoA to study the similarity or divergence of sample community composition. Based on values generated using the weighted UniFrac set of rules, the PCoA confirmed that the intestinal microbiota of the ABx group became separated from the NC group (P<0.05), suggesting that the gut microbiota structure in ABx mice significantly differed from the NC group (Fig 4B and 4C). After 11 days, the gut microbiota of the M group became nonetheless separated from the NC group (P<0.05), and the distance between the TL, TH, and NC groups was closer than the between M and NC groups (P<0.05). These results suggested that the structure of the intestinal flora of TB-treated mice was closer to the NC group (Fig 4E and 4F).

**TB altered the gut microbiota composition in antibiotic-induced mice.** On day seven, at the phylum level, the antibiotic treatment decreased the relative abundance of *Bacteroidetes* but expanded the relative affluence of *Firmicutes*, *Actinobacteria*, and *Proteobacteria* compared with the NC group. At the genus level, antibiotic treatment reduced the relative abundance of *Muribaculaceae* but increased the relative abundance of *Staphylococcus* compared to the NC group (Fig 5A–5E).

On day 18, at the phylum level, *Bacteroidetes*, *Firmicutes*, *Actinobacteria* and *Proteobacteria* were the main phyla within the fecal microflora (Fig 5F). At the genus level, compared to the NC group, the relative abundance of *Muribaculaceae* in the M group appreciably decreased, the relative abundance of *Bacteroides* and *Enterococcus* increased. After TB intervention, the relative abundance of *Muribaculaceae* and *Bifidobacterium* in the TL and TH groups was significantly up-regulated, and the relative abundance of the TL group was higher than that of the TH group. In addition, the relative abundance of *Bacteroides* and *Enterococcus* decreased significantly, and the relative abundance of the TL group was lower than that of the TH group (Fig 5G–5K).

Then, we used the LDA Effect Size to investigate each group of signature genus. *Clostridium_sensu_stricto1* was enriched in M mice. *Muribaculaceae*, *Bifidobacterium*, *Parabacteroides*, and *Flavonifractor* were dominant in TL mice, and *Ruminiclostridium 5*, *Alistipes*, and *Desulfovibrio* were enriched in TH mice (Fig 5L and 5M).

## TB increased SCFAs content in feces

After seven days of antibiotics treatment, acetic, propionic, isobutyric, butyric, and valeric acids significantly decreased in the ABx group compared to the NC group (P<0.05) (Fig 6A–6F). Hexanoic acid presented a similar trend but was not statistically significant. On day 18, the contents of acetic and propionic acid in the M group were significantly lower than those in the NC group, and the contents of acetic, propionic and butyric acid in the TL group were significantly higher than those in the M group (P<0.05), the contents of acetic, propionic and butyric acid in the TH group were significantly lower than those in the TL group (P<0.05) (Fig 6G–6L).

## TB inhibited the activation of *NLRP3* inflammasome and improved intestinal inflammation

When gut microbiota is disturbed, the *NLRP3* inflammasome can be activated by the production of large LPS amounts by Gram-negative bacteria, causing an inflammatory response [29].

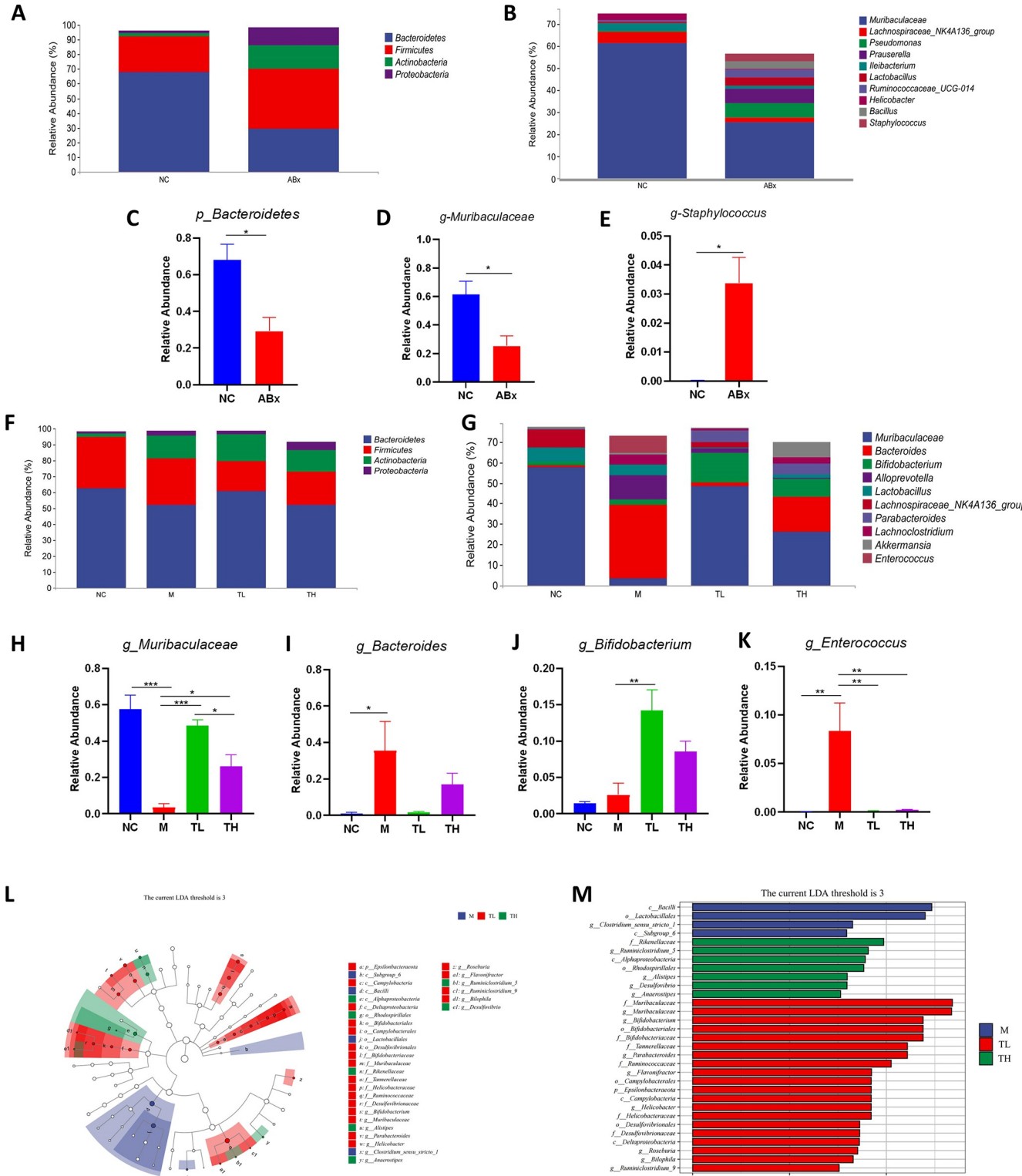

**Fig 5. TB altered the composition of gut microbiota in antibiotic-induced mice.** (**A**) On day 7, Relative abundances of bacterial phylum level, (**B**) On day 7, Relative abundances of bacterial genus level, (**C**) Levels of *Bacteroidetes*, (**D**) Levels of *Muribaculaceae*, (**E**) Levels of *staphylococcus*, (**F**) On day 18, Relative abundances of bacterial phylum level, (**G**) On day 18, Relative abundances of bacterial genus level, (**H**) Levels of *Muribaculaceae*, (**I**) Levels of *Bacteroides*, (**J**) Levels of *Bifidobacterium*, (**K**) Levels of *Enterococcus*, (**L**) Cladogram illustrating the results of LEfSe analysis, (**M**) LDA scores for bacterial taxa significantly enriched in gut microbiota from each group (LDA score > 3). NC: control group; ABx: ABx group; M: model group; TL: TB low dose group; TH: TB high dose group. *P<0.05; ** P<0.01; ***P<0.001.

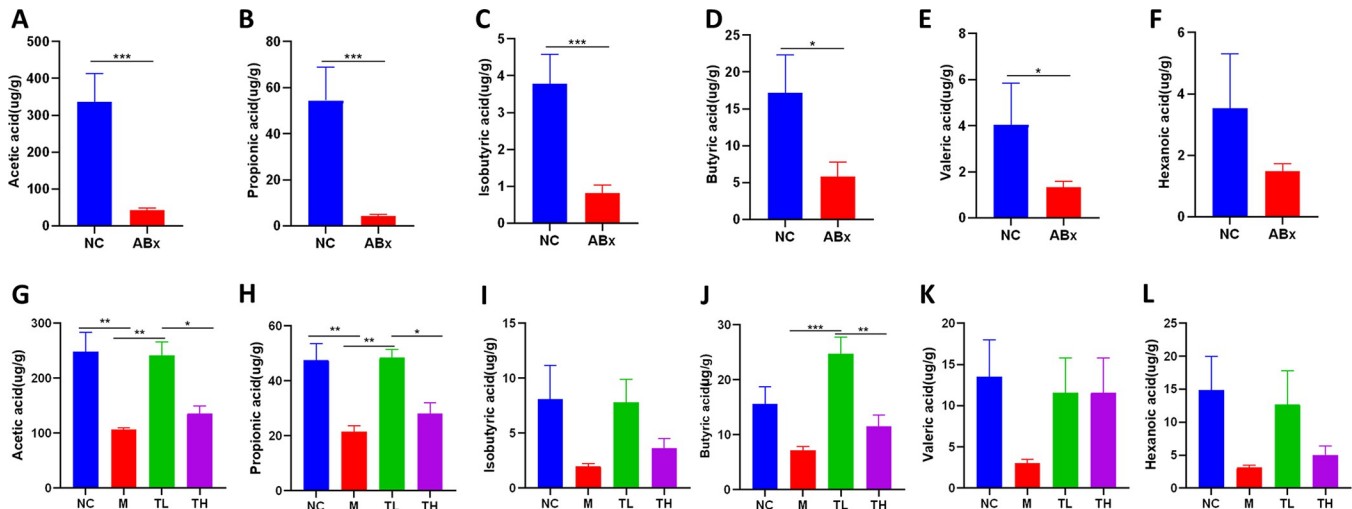

**Fig 6. TB increased SCFAs content in feces.** On day 7: **(A)** Acetic acid, **(B)** Propionic acid, **(C)** Isobutyric acid, **(D)** Butyric acid, **(E)** Valeric acid and **(F)** Hexanoic acid; On day 18: **(G)** Acetic acid, **(H)** Propionic acid, **(I)** Isobutyric acid, **(J)** Butyric acid, **(K)** Valeric acid and **(L)** Hexanoic acid; NC: control group; M: model group; TL: TB low dose group; TH: TB high dose group. *P<0.05; ** P<0.01; ***P<0.001.

Therefore, we detected the mRNA and protein levels of *NLRP3* inflammasome-associated genes and inflammatory elements *TNF-α* and *IL-6* in the colon tissues of mice. Compared to NC mice, the mRNA levels of *NLRP3*, *ASC*, *caspase-1*, *IL-1β*, *TNF-α*, and *IL-6* significantly increased in the M group (P<0.05) (Fig 7A–7F). After TB intervention, mRNA expression levels of the above factors in the TL group were significantly decreased compared with that in the M group, and mRNA expression levels of *IL-6*, *NLRP3*, *ASC*, *caspase-1* and *IL-1β* in the TL group were significantly lower than those in the TH group (P< 0.05).

The levels of *NLRP3* inflammasome-related proteins in colon tissues were also detected by Western blot. The protein expressions of *NLRP3*, *ASC*, *caspase-1* and *IL-1β* in colon tissue of mice in the M group were significantly increased compared with those in the NC group (P<0.05), indicating that the *NLRP3* inflammasome is activated. After TB intervention, compared with the M group, the expression of above proteins in colon tissues of the TL group was significantly down-regulated (P<0.05), the protein expressions of *NLRP3*, *ASC* and *caspase-1* in TL group were significantly lower than those in the TH group (P<0.05) (Fig 7G–7K). The above results were consistent with those of PCR experiment, indicating that low dose TB intervention could inhibit the overactivation of *NLRP3* inflammasome and thus reduce intestinal inflammatory response.

## TB improved the integrity of the intestinal barrier in antibiotic-induced mice

Alcian blue stains the nucleus of intestinal mucosal epithelial cells red and the mucous in the cytoplasm of goblet cells blue. The intestinal goblet cells of NC mice were abundant and neatly arranged. In contrast, goblet cells in the M group were depleted and disordered. After TB intervention, intestinal goblet cells significantly increased in the TL group, which was not evident in the TH group. Consistent with these changes, the mRNA levels of *MUC2*, related to the synthesis and secretion capability of goblet cells, significantly decreased in the colons of M mice than in NC mice (P<0.05). *MUC2* expression increased in the colons of TL and TH mice. Low-dose TB increased *MUC2* expression more effectively compared to the M group (P<0.05) (Fig 8B). We subsequently analyzed epithelial cell integrity by inspecting the mRNA

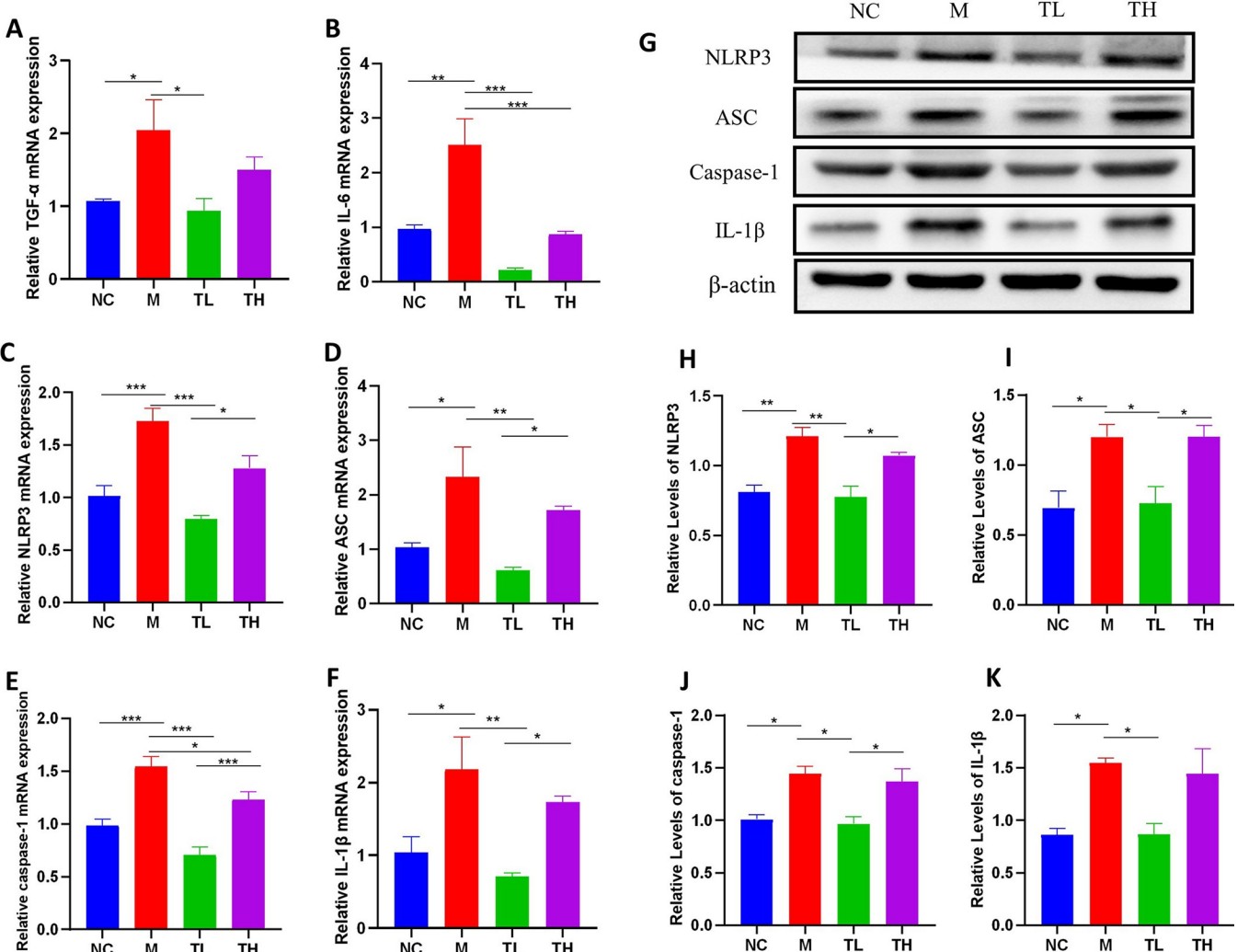

**Fig 7. TB inhibited the activation of *NLRP3* inflammasome and improved intestinal inflammation. (A)** *TNF-α* mRNA in colon, **(B)** *IL-6* mRNA in colon, **(C)** *NLRP3* mRNA in colon, **(D)** *ASC* mRNA in colon, **(E)** *caspase-1* mRNA in colon, **(F)** *IL-1β*mRNA in colon. **(G)** Expression of *NLRP3, ASC, caspase-1* and *IL-1β* protein in colon. **(H)** *NLRP3* protein in colon, **(I)** *ASC* protein in colon, **(J)** *caspase-1* protein in colon, **(K)** *IL-1β* protein in colon. NC: control group; M: model group; TL: TB low dose group; TH: TH high dose group. *P<0.05; ** P<0.01; ***P<0.001.

and protein levels of tight junction elements. The mRNA and protein levels of *ZO-1* and *Occludin* in the M group decreased compared to the NC group (P<0.05), which increased after low-dose TB treatment (P<0.05), and the protein level of *Occludin* in TL group was significantly higher than that in TH group (P<0.05), (Fig 8E–8I). The destruction of the intestinal barrier might lead to increased intestinal permeability and luminal pathogens' entrance into the bloodstream. Compared with the NC group, serum levels of LPS and zonulin in the M group were significantly increased (P<0.05), after TB intervention, LPS and zonulin levels in the TL and TH groups were significantly decreased compared with those in the M group (P<0.05), and the level of TL group was significantly lower than TH group (P<0.05) (Fig 8C and 8D).

## Discussion

Gut microbiota disorder appears in many diseases, such as gastrointestinal, metabolic, neuro-degenerative, and cardiovascular diseases [3, 30–33]. Herein, we used the antibiotic-induced

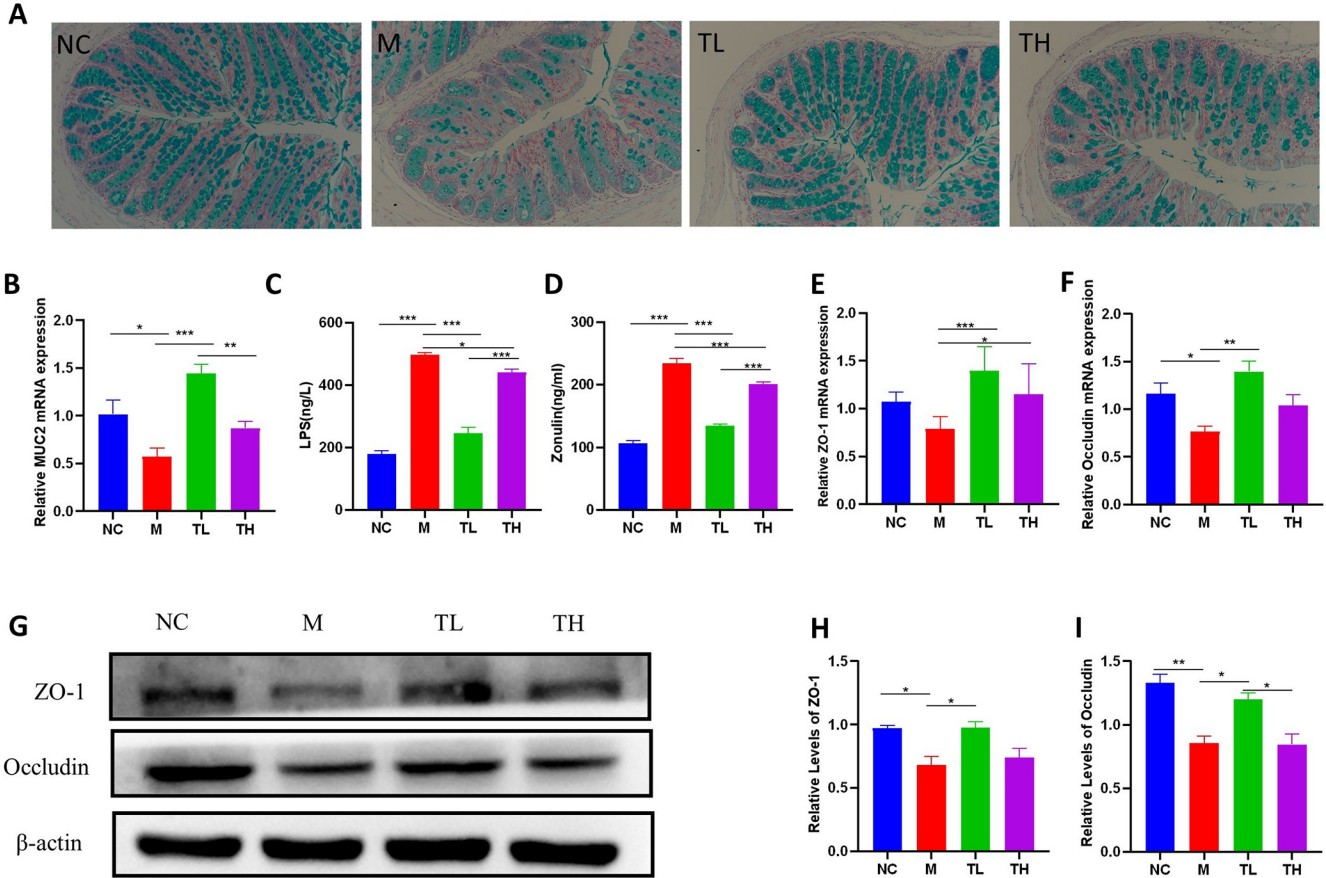

**Fig 8. TB improved the integrity of the intestinal barrier in antibiotic-induced mice.** (A) Alcian blue staining, 400×; (B) *MUC2* mRNA in colon, (C) serum lipopolysaccharide (LPS); (D) serum zonulin; (E) *ZO-1* mRNA in colon, (F) *Occludin* mRNA in colon. (G) Expression of *ZO-1* and *Occludin* protein in colon. (H) *ZO-1* protein in colon, (I) *Occludin* protein in colon. NC: control group; M: model group; TL: TB low dose group; TH: TH high dose group. *P<0.05; ** P<0.01; ***P<0.001.

gut microbiota disorder model to explore the influence of TB, the precursor of butyric acid, on this disorder.

The most prominent feature of gut microbiota disorder is the loss of species diversity [34]. We found that, after the antibiotic intervention, the α diversity significantly decreased in mice, and weight loss, diarrhea, increased cecal contents, and colon length were observed in all mice. The gut microbiota of mice is disturbed after antibiotic intervention [35]. As the diversity and abundance of bacteria decrease, the mice's gut digestion slows down [36], so they eat less and lose weight. Studies have shown that intestinal peristalsis slows down after gut microbiota reduction, intestinal excretion decreases, and the liquid stays in the intestine, increasing cecal contents and volume [37]. Additionally, diarrhea is a common adverse reaction to antibiotics [38]. Consistently, we found diarrhea symptoms in mice treated with antibiotics. Previous studies have suggested that antibiotic diarrhea might be caused by allergic and toxic effects of antibiotics on the intestinal mucosa and disturbances in the composition and function of the normal gut microbiome [39]. After TB intervention, these symptoms can be improved, and low-dose TB was better than high-dose.

We also found that TB had a positive regulatory effect on antibiotic-induced gut microbiota disorder, mainly manifested as changes in structure and composition. After TB intervention, the composition of gut microbiota mice was more similar to the NC group. Low-dose TB

increased the relative abundance of *Muribaculaceae*, *Bifidobacterium* and *Parabacteroides* and reduced *Bacteroidetes*, *Alloprevotella*, and *Enterococcus*. High-dose TB did not work as well as low-dose TB. Studies have shown that the *Muribaculaceae* abundance strongly correlates with propionic acid [40]. However, propionic acid inhibits CD8+ T cell activation, which is poorly correlated with the prevalence of inflammation [41]. *Bifidobacterium* is a well-known beneficial bacteria in the intestine of humans and animals, as a butyrate-producing genus recognised to play a protecting position within the human intestine barrier by protecting against pathogens and diseases [42]. There is growing evidence that *Bifidobacterium* acts as a probiotic to prevent and relieve diarrhea and IBD [43]. A clinical study has shown that *Bifidobacteria* intervention reduces inflammation levels in ulcerative colitis patients [44]. *Parabacteroides* have the physiological characteristics of metabolizing carbohydrates and producing SCFAs [45]. One study showed that *Parabacteroides* modulate inflammatory markers and promote intestinal barrier integrity, attenuating tumorigenesis in azomethane-treated A/J mice [46]. Conversely, the colonization of some pathogenic bacteria can promote enteritis development. *Bacteroides* species sometimes play an important role in the human metabolic system but can also lead to diseases [47]. LPS, as one of the main products of *Bacteroides*, can cause an intestinal epithelial inflammatory response and barrier dysfunction [48]. One study suggested that chlorogenic acid protects intestinal integrity and alleviates intestinal inflammation by inhibiting *Bacteroidetes*' growth and reducing LPS produced by Bacteroidetes [48]. *Alloprevotella* and *Enterococcus* are potential harmful bacteria. For example, *Alloprevotella* can be more abundant in AOM/DSS-treated mice [49]. *Enterococcus* induce inflammatory diseases including IBD and hepatic inflammation [50]. These changes indicated that after TB intervention, gut microbiota disorder was restored, increasing beneficial and decreasing harmful bacteria abundance.

TB is a precursor to butyric acid, the main SCFAs, which has many positive effects on health [51]. Our results showed that TB supplementation could increase butyric acid and other SCFAs, such as acetic and propionic acids. Low-dose TB had an advantage over high-dose. Therefore, TB has a restorative effect on SCFAs, possibly because TB supplementation enables butyric acid to reach the intestine. Another reason is that TB stimulates the increase of SCFAs-producing bacteria (such as *Muribaculaceae*, *Bifidobacterium*, and *Parabacteroides*), increasing the content of other SCFAs.

Our results indicated that the levels of serum LPS and colonic inflammatory factors (IL-6 and TNF-α) were significantly higher in M mice than in NC mice, suggesting that M mice were in a low systemic inflammation state. Low-dose TB supplementation significantly down-regulated serum LPS levels and colonic inflammatory cytokines (*IL-6* and *TNF-α*) in antibiotic-induced mice, while high-dose TB supplementation did not have the same effect. Studies have reported that inflammatory cytokines and LPS can activate the *NLRP3/ASC/Caspase-1* signaling pathway and produce the pro-inflammatory cytokine *IL-1β* [29, 52]. In addition, other studies have shown that activation of *NLRP3* inflammasome may be key to triggering intestinal inflammation [53]. Herein, compared with the NC group, the *NLPR3*, *ASC*, *caspase-1*, and *IL-1β* levels increased in the colon of M mice, indicating that the *NLRP3/ASC/Caspase-1* signaling pathway was activated. After TB treatment, the expressions of *NLRP3* inflammasome-related factors and *IL-1β* in the colon of TL mice significantly decreased. The TH group did not show the same effect. Many studies have shown that SCFAs inhibit LPS-induced inflammation [54–56]. Therefore, we hypothesized that low-dose TB treatment increased the abundance of SCFAs-producing bacteria, thereby increasing SCFAs contents, inhibiting LPS-induced NLRP3 inflammasome activation, and alleviating the inflammatory response.

Inflammation of the colon is usually accompanied by an intestinal barrier breakdown, manifested by a massive depletion of intestinal goblet cells and a breakdown of tight junction (TJ). In this study, TB intervention significantly increased the number of intestinal goblet cells and

the expression of *MUC2*. Additionally, the levels of tight junction elements (*ZO-1* and *Occludin*) were significantly increased, and the low dose's effects were better than the high dose. Recent studies have shown that physiological concentrations of *IL-1β* and *TNF-α* significantly increase intestinal epithelial TJ permeability [57], and inhibiting the *IL-1β*-induced increase in intestinal TJ permeability prevents sodium dextran sulfate (DSS)-induced intestinal inflammation [58]. Therefore, low-dose TB intervention might inhibit the expression of intestinal *IL-1β* and protect intestinal tight junctions to maintain intestinal barrier integrity.

In summary, we demonstrated the benefits of TB supplementation on gut microbiota disorder caused by antibiotics. Our results revealed that TB might restore gut microbiota to produce SCFAs, inhibit the over-activation of the *NLRP3* inflammasome and attenuate intestinal injury.

## Supporting information

**S1 Dataset.**
(XLSX)

**S1 Raw images.**
(PDF)

## Acknowledgments

The authors gratefully acknowledge all team members for her contributions to this article.

## Author Contributions

**Conceptualization:** Ning Yang, Tongtong Lan, Yang Song.

**Data curation:** Ning Yang, Yisa Han, Haifeng Zhao.

**Formal analysis:** Chuhui Wang, Zhen Xu, Meng Tao, Hui Li.

**Methodology:** Ning Yang, Tongtong Lan, Yang Song.

**Supervision:** Xuezhen Ma.

**Validation:** Ning Yang.

**Visualization:** Ning Yang, Yisa Han, Zhao Chen.

**Writing – original draft:** Ning Yang.

**Writing – review & editing:** Tongtong Lan, Yang Song.

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
