## [Decision Letter · Decision Letter 0]

31 May 2023

PONE-D-23-04467Tributyrin alleviates gut microbiota dysbiosis to repair intestinal damage in antibiotic-treated micePLOS ONE

Dear Dr. Song,

Thank you for submitting your manuscript to PLOS ONE. After careful consideration, we feel that it has merit but does not fully meet PLOS ONE’s publication criteria as it currently stands. Therefore, we invite you to submit a revised version of the manuscript that addresses the points raised during the review process.

We look forward to receiving your revised manuscript.

Kind regards,

Marwa Ibrahim Abd El-Hamid

Academic Editor

PLOS ONE

Journal Requirements:

a) The name of the colleague or the details of the professional service that edited your manuscript.

b) A copy of your manuscript showing your changes by either highlighting them or using track changes (uploaded as a *supporting information* file).

c) A clean copy of the edited manuscript (uploaded as the new *manuscript* file).

"Y.S.was supported by the National Natural Science Foundation of China (81973033)."

**Additional Editor Comments:**

Dear authors

An English edition is needed as per the reviewer comment.

Reviewers' comments:

Reviewer's Responses to Questions

**Comments to the Author**

1. Is the manuscript technically sound, and do the data support the conclusions?

Reviewer #1: Yes

2. Has the statistical analysis been performed appropriately and rigorously? 

Reviewer #1: Yes

3. Have the authors made all data underlying the findings in their manuscript fully available?

Reviewer #1: Yes

4. Is the manuscript presented in an intelligible fashion and written in standard English?

Reviewer #1: Yes

5. Review Comments to the Author

Reviewer #1: The authors established an intestinal microbiota disorder model using ceftriaxone sodium via gavage to investigate the effects of different TB doses for restoring gut microbiota and intestinal injury. It sound excellent.

1.Many careless mistakes are involved in the manuscript, eg. In Abstract, writing as ABx group or Abx group. In the manuscript, writing as SCFAs or SCFA, muc2 or MUC2; p or P; ‘. [35].’；“Studies have shown that the Muribaculaceae abundance strongly correlates with propionic acid.”this sentence needs to supported by references. Some bacteria are italic, but others not.

2.Please tell the reason for 11 days of TB interventions, the reason for using daily dose of 0.3 and 3g/kg, respectively, not for other concentrations.

3.In Methods section, how to use GC/MSD analysis to measure SCFAs needs to be described.

4.LPS level was determined by ELISA? ELISA is based on antibody and antigen response, but LPS determination based on Limulus reagent, it’s different.

5.In Methods section, primary and secondary antibodies used in Western blot needs to provide the products numbers and associated companies. And all primary antibodies are‘Rabbit polyclonal antibody’, is that true? Why not use monoclonal antibody?

6.Statistical analysis: compared by analysis of variance (ANOVA), here, not complete, should like one-way or two-ways...; why just use nonparametric test, for the mice, represents a relative consistent.

7.In figure 5, Bifidobacterium in genus level is increased with TB treatment, but normal level in NC group is too low as similar as M group, how to explain? In addition, several bacterial showed no difference, such as Alloprevotella and Parabacteroides, why the authors showed this in figure 5?

8.My another concern is the model. ceftriaxone sodium was used in this study to generate Model for gut dysbiosis, but this antibiotics is broad-spectrum anti-bacteria including anti-probiotics and anti-pathogens, the authors should provide single differential bacteria figure in Figure 5 between NC and Abx groups to help us know the specific components features of gut dysbiosis generated by ceftriaxone sodium. In figure 5, the differential bacteria analysis between NC and Abx groups needs to be shown one by one.

9.Many grammar errors needs to be corrected.

Such as: several sentences needs to be rectified: After TB intervention, these indexes were reduced compared to M mice, and the TL group was better than the TH group. Rectify this sentence.

Similarly, please rectify “After TB intervention, these indicators were reduced compared to the M group, and the TL group was better than the TH group (p < 0.05).”

‘the mice's gut digestion slows down, so they eat less and lose weight.’

10.Which inflammatory cells are responsible for the inflammation. The authors needs to detect this.

11.As we known, TB or butyrate suppress the inflammation via binding GPR receptors or inhibiting HDACs, but these molecular pathways are not presented in the manuscript.

6. PLOS authors have the option to publish the peer review history of their article (what does this mean?). If published, this will include your full peer review and any attached files.

Reviewer #1: No

---

## [Author Response · Author response to Decision Letter 0]

14 Jul 2023

Dear Editor and Reviewers:

We thank the reviewers for carefully reading our paper "Tributyrin alleviates gut microbiota dysbiosis to repair intestinal damage in antibiotic-treated mice", and for giving us excellent comments. According to your suggestion, we have revised the manuscript and marked the revised parts with yellow background color. Attached you will find our point-by-point responses to the comments.

We look forward to your favorable decision and thank you for the opportunity of allowing us to revise this paper.

1.Many careless mistakes are involved in the manuscript, eg. In Abstract, writing as ABx group or Abx group. In the manuscript, writing as SCFAs or SCFA, muc2 or MUC2; p or P; ‘. [35].’ “Studies have shown that the Muribaculaceae abundance strongly correlates with propionic acid.” this sentence needs to supported by references. Some bacteria are italic, but others not.

Response: Thank you very much for your comments. Meanwhile, we are very sorry for the inconvenience caused by our mistake. According to the reviewer's suggestion, we have revised "Abx" to "ABx "," SCFA "to" SCFAs ", "muc2" to "MUC2", "p to P", changed the bacterial names in italics, and inserted references where the reviewer suggested(L321). 

2.Please tell the reason for 11 days of TB interventions, the reason for using daily dose of 0.3 and 3g/kg, respectively, not for other concentrations.

Response: Thank you very much for pointing out the valuable comment. Our study focused on the repair effect of TB short-term intervention on gut microbiota disorders in mice. Studies have shown that the self-repair cycle of the gut flora is about 14 days[1]. And many studies exploring the short-term intervention for repairing antibiotic-induced gut microbiota disorders in mice have mostly lasted for 10-14 days [2-4], so we chose to intervene for 11 days. Relevant references have been inserted in the experimental design section (L84）. 

It has been documented in the literature that TB can play a good inhibitory effect on colorectal cancer at the dose of 200 mg/100 g in rats [5](the dose is converted to about 3g/kg in mice), and play a good inhibitory effect on colitis at the dose of 3g/kg in mice[6]. Therefore, we took this dose as the high dose TH for this experiment, and set a low dose group downward[7].

3.In Methods section, how to use GC/MSD analysis to measure SCFAs needs to be described.

Response: Thanks for your kind suggestion. We have added GC/MSD conditions methods to the “Methods” section(L116-123).

The chromatographic conditions were as follows: the inlet temperature was 230℃, the carrier gas was high purity helium, and the column flow rate was 1.0 mL/min (constant current mode). The initial temperature was 70℃ for 1 minute, and the temperature was heated to 110℃ at 20℃ /min, and then heated to 230°C at 10°C /min at 180℃ at 5℃ /min for 6 minutes. The mass spectrum conditions are as follows: the ionization mode is the electron bombardment ion source (EI), the electron bombardment energy is 70 eV, the ion source temperature is 230℃, and the electron multiplier voltage is the automatic tuning value. The scanning mode of mass spectrum data is selective ion scanning (SIM). Selected ions: 41, 42, 43, 45, 55, 57, 60, 73, 74, 87, 88, solvent delay time was 4.2 minutes. 

4.LPS level was determined by ELISA? ELISA is based on antibody and antigen response, but LPS determination based on Limulus reagent, it’s different.

Response: We thank the reviewer for pointing out the comment. At present, the main methods for detecting LPS include Limulus reagent and ELISA[8-10]. Limulus reagent is the gold standard for endotoxin detection because of its rapid, simple, sensitive and reproducible characteristics. But the preparation of limulus requires the killing of a large number of protected animals. As a substitute for limulus method, ELISA has the advantages of good specificity and high accuracy. Therefore, in this study, ELISA was used to detect serum LPS levels.

5.In Methods section, primary and secondary antibodies used in Western blot needs to provide the products numbers and associated companies. And all primary antibodies are‘Rabbit polyclonal antibody’, is that true? Why not use monoclonal antibody?

Response: Thanks to the reviewer for reminding us of this detail. We have added the products numbers and related companies of primary and secondary antibodies in the Methods section. In our study, we did use Rabbit polyclonal antibody for Western blot. Because polyclonal antibodies can improve the sensitivity of detection and are easier to detect proteins with low abundance, in addition, the cost of polyclonal antibodies is also lower than that of monoclonal antibodies. To save time and cost, therefore, we chose polyclonal antibodies in the Western blot experiment.

6.Statistical analysis: compared by analysis of variance (ANOVA), here, not complete, should like one-way or two-ways...; why just use nonparametric test, for the mice, represents a relative consistent.

Response: We thank the reviewer for reminding us that detail. We have added variance (ANOVA) as a One-Way Analysis of Variance (One-Way ANOVA).

In LEfSe analysis, Kruskal-Wallis test was used to detect species with significant abundance differences among different groups[11].

7.In figure 5, Bifidobacterium in genus level is increased with TB treatment, but normal level in NC group is too low as similar as M group, how to explain? In addition, several bacterial showed no difference, such as Alloprevotella and Parabacteroides, why the authors showed this in figure 5?

Response: We thank the reviewer for the valuable comment. After the modeling, there was no significant difference in the relative abundance of Bifidobacterium between NC group and ABx group, indicating that ceftriaxone sodium modeling had not reduced the relative abundance of Bifidobacterium. Therefore, on the 18th day, there was no significant difference in the relative abundance of Bifidobacterium between the M and NC groups. However, the addition of TB has significantly increased the relative abundance of Bifidobacterium.

Genera with no statistical difference have been removed on expert advice.

8. My another concern is the model. ceftriaxone sodium was used in this study to generate Model for gut dysbiosis, but this antibiotics is broad-spectrum anti-bacteria including anti-probiotics and anti-pathogens, the authors should provide single differential bacteria figure in Figure 5 between NC and Abx groups to help us know the specific components features of gut dysbiosis generated by ceftriaxone sodium. In figure 5, the differential bacteria analysis between NC and Abx groups needs to be shown one by one.

Response: Thank you for your professional advice. We have shown the difference taxa between the NC group and Abx group one by one according to your suggestion in Figure 5C, D, and E.

9.Many grammar errors needs to be corrected.

Such as: several sentences needs to be rectified: After TB intervention, these indexes were reduced compared to M mice, and the TL group was better than the TH group. Rectify this sentence.

Response: Thank you for your careful review. Meanwhile, we sincerely apologize for the inconvenience caused by our mistake. 

Correction was done as follows:

L256-L259: “After TB intervention, these indexes were reduced compared to M mice, and the TL group was better than the TH group” → “After TB intervention, mRNA expression levels of the above factors in the TL group were significantly decreased compared with that in the M group, and mRNA expression levels of IL-6, NLRP3, ASC, caspase-1 and IL-1β in the TL group were significantly lower than those in the TH group (P< 0.05)”

Language, especially the grammar and tense, and the format layout were corrected in the revised manuscript.

10.Which inflammatory cells are responsible for the inflammation. The authors needs to detect this.

Response: Thank you for your professional advice. A large number of literatures have shown that TNF-α and IL-6 are typical inflammatory factors, mainly produced by macrophages and monocytes[12, 13], and their high expression can indicate the inflammatory response of tissues. In this study, the purpose of our research is mainly to illustrate the inhibitory effect of TB on intestinal inflammation. Which inflammatory cells produce inflammation can be further studied in the future.

11.As we known, TB or butyrate suppress the inflammation via binding GPR receptors or inhibiting HDACs, but these molecular pathways are not presented in the manuscript.

Response: We thank the reviewer for the valuable comment. Indeed, the classic pathway for butyrate to inhibit inflammation is to inhibit inflammation by binding to GPR receptors or inhibiting HDAC. However, in this study, we focused on the intestinal flora, exploring that TB inhibits inflammation by repairing the imbalance of intestinal flora and down-regulating LPS to inhibit the over-activation of NLRP3 inflammasome, so GPR receptor and HDAC detection were not performed.

Reference

1. Jialing L, Yangyang G, Jing Z, Xiaoyi T, Ping W, Liwei S, et al. Changes in serum inflammatory cytokine levels and intestinal flora in a self-healing dextran sodium sulfate-induced ulcerative colitis murine model. Life Sci. 2020;263:118587.

2. Shi T, Bian X, Yao Z, Wang Y, Gao W, Guo C. Quercetin improves gut dysbiosis in antibiotic-treated mice. Food Funct. 2020;11(9):8003-8013.

3. Bian X, Shi T, Wang Y, Ma Y, Yu Y, Gao W, et al. Gut dysbiosis induced by antibiotics is improved by tangerine pith extract in mice. Nutr Res. 2022;101:1-13.

4. Bai X, Fu R, Duan Z, Wang P, Zhu C, Fan D. Ginsenoside Rk3 alleviates gut microbiota dysbiosis and colonic inflammation in antibiotic-treated mice. Food Res Int. 2021;146:110465.

5. Heidor R, Furtado KS, Ortega JF, de Oliveira TF, Tavares PE, Vieira A, et al. The chemopreventive activity of the histone deacetylase inhibitor tributyrin in colon carcinogenesis involves the induction of apoptosis and reduction of DNA damage. Toxicol Appl Pharmacol. 2014;276(2):129-135.

6. Fachi JL, Felipe JS, Pral LP, da Silva BK, Correa RO, de Andrade MCP, et al. Butyrate Protects Mice from Clostridium difficile-Induced Colitis through an HIF-1-Dependent Mechanism. Cell Rep. 2019;27(3):750-761 e757.

7. Meng Y LX, Zhao XM,et al. Effect of Butyric acid on immune function in mice with colorectal cancer liver metastasis implanted in situ. Acta Nutrimenta Sinica. 2019; 2019，41(2)：163-167.

8. Martinez-Sernandez V, Orbegozo-Medina RA, Romaris F, Paniagua E, Ubeira FM. Usefulness of ELISA Methods for Assessing LPS Interactions with Proteins and Peptides. PLoS One. 2016;11(6):e0156530.

9. Jiang Z, Li L, Liu L, Ding B, Yang Y, He F, et al. Ischemic Stroke and Dysbiosis of Gut Microbiota: Changes to LPS Levels and Effects on Functional Outcomes. Altern Ther Health Med. 2023;29(5):284-292.

10. Lu M, Sun J, Zhao Y, Zhang H, Li X, Zhou J, et al. Prevention of High-Fat Diet-Induced Hypercholesterolemia by Lactobacillus reuteri Fn041 Through Promoting Cholesterol and Bile Salt Excretion and Intestinal Mucosal Barrier Functions. Frontiers in nutrition. 2022;9:851541.

11. Chang F, He S, Dang C. Assisted Selection of Biomarkers by Linear Discriminant Analysis Effect Size (LEfSe) in Microbiome Data. J Vis Exp. 2022;(183).

12. Tanaka T, Narazaki M, Masuda K, Kishimoto T. Regulation of IL-6 in Immunity and Diseases. Adv Exp Med Biol. 2016;941:79-88.

13. Idriss HT, Naismith JH. TNF alpha and the TNF receptor superfamily: structure-function relationship(s). Microsc Res Tech. 2000;50(3):184-195.

---

## [Editor Report · Decision Letter 1]

18 Jul 2023

Tributyrin alleviates gut microbiota dysbiosis to repair intestinal damage in antibiotic-treated mice

PONE-D-23-04467R1

Dear Dr. Song,

We’re pleased to inform you that your manuscript has been judged scientifically suitable for publication and will be formally accepted for publication once it meets all outstanding technical requirements.

Kind regards,

Marwa Ibrahim Abd El-Hamid

Academic Editor

PLOS ONE
---

## [Editor Report · Acceptance letter]

21 Jul 2023

PONE-D-23-04467R1 

Tributyrin alleviates gut microbiota dysbiosis to repair intestinal damage in antibiotic-treated mice 

Dear Dr. Song:

I'm pleased to inform you that your manuscript has been deemed suitable for publication in PLOS ONE. Congratulations! Your manuscript is now with our production department. 

Kind regards, 

on behalf of

Dr. Marwa Ibrahim Abd El-Hamid 

Academic Editor

PLOS ONE